# Dual Mode of Action of Acetylcholine on Cytosolic Calcium Oscillations in Pancreatic Beta and Acinar Cells In Situ

**DOI:** 10.3390/cells10071580

**Published:** 2021-06-23

**Authors:** Nastja Sluga, Sandra Postić, Srdjan Sarikas, Ya-Chi Huang, Andraž Stožer, Marjan Slak Rupnik

**Affiliations:** 1Faculty of Medicine, University of Maribor, 2000 Maribor, Slovenia; nastja.sluga1@um.si (N.S.); andraz.stozer@um.si (A.S.); 2Center for Physiology and Pharmacology, Medical University of Vienna, 1090 Vienna, Austria; sandra.postic@meduniwien.ac.at (S.P.); srdjan.sarikas@meduniwien.ac.at (S.S.); yachi.huang@utoronto.ca (Y.-C.H.); 3Alma Mater Europaea, European Center Maribor, 2000 Maribor, Slovenia

**Keywords:** pancreas tissue slices, acetylcholine, beta cell, acinar cell, Ca^2+^ oscillations

## Abstract

Cholinergic innervation in the pancreas controls both the release of digestive enzymes to support the intestinal digestion and absorption, as well as insulin release to promote nutrient use in the cells of the body. The effects of muscarinic receptor stimulation are described in detail for endocrine beta cells and exocrine acinar cells separately. Here we describe morphological and functional criteria to separate these two cell types in situ in tissue slices and simultaneously measure their response to ACh stimulation on cytosolic Ca^2+^ oscillations [Ca^2+^]_c_ in stimulatory glucose conditions. Our results show that both cell types respond to glucose directly in the concentration range compatible with the glucose transporters they express. The physiological ACh concentration increases the frequency of glucose stimulated [Ca^2+^]_c_ oscillations in both cell types and synchronizes [Ca^2+^]_c_ oscillations in acinar cells. The supraphysiological ACh concentration further increases the oscillation frequency on the level of individual beta cells, inhibits the synchronization between these cells, and abolishes oscillatory activity in acinar cells. We discuss possible mechanisms leading to the observed phenomena.

## 1. Introduction

The pancreatic endocrine islet cells and exocrine acinar and ductal cells are richly innervated by parasympathetic, sympathetic, and sensory nerves [1,2]. Cholinergic nerve fibers entering the pancreas are predominantly preganglionic. These fibers originate in the dorsal motor nucleus of N. vagus [3,4] and are under hypothalamic control [5]. Preganglionic fibers terminate in intrapancreatic ganglia. Postganglionic nerves emanate from these ganglia to terminate in the vicinity of hormone and digestive enzyme secreting cells [6]. In mice, postganglionic nerve fibers communicate with all types of endocrine cells in the islets. In contrast, parasympathetic innervation is rather scarce in exocrine tissue [7]. Furthermore, a considerably higher concentration of choline acetyltransferase is found in islets compared to exocrine tissue [8].

Preganglionic parasympathetic nerve fibers release acetylcholine (ACh) that binds to nicotinic receptors in intraganglionic neurons [1]. Inhibition of these receptors has been found to abolish insulin secretion in beta cells [9], suggesting the importance of vagal descending pathways. ACh released by postganglionic fibers through the activation of the muscarinic receptors directly triggers insulin release from islet beta cells [10] and mediates digestive enzyme secretion in acinar cells [11]. It is noteworthy that the effects of different neurotransmitters released by vagal fibers on beta cell insulin secretion differs between species [2]. Similarly, intraganglionic innervation is species-dependent [12,13], demonstrating different neuronal regulation of pancreatic endocrine function between species [13]. In both mice and humans, parasympathetic nerve endings that innervate pancreatic islets release ACh. Interestingly, human alpha cells express vesicular acetylcholine transporter and release ACh in a paracrine manner to support optimal response of beta cells to glucose [14], representing a non-neuronal source of ACh in the cholinergic system in the human islets [15].

Pharmacological characterization revealed that in mice both beta and acinar cells express the muscarinic acetylcholine receptor subtype 1 (M_1_R) and 3 (M_3_R) (Figure 1) [16]. Human beta cells additionally express M_5_ receptors [17]. Glucose-induced insulin release has been demonstrated to be mediated through the activation of M_3_R [16,18]. Mice selectively lacking M_3_R in beta cells show impaired glucose tolerance and reduced insulin release, whereas mice selectively overexpressing M_3_R in pancreatic beta cells show improved glucose tolerance and increased insulin release [19]. The activation of parasympathetic nerve fibers and ACh release is not a transient event and does not occur only in the preabsorptive phase of feeding, but persists throughout the absorptive phase [19]. The cholinergic stimulation in pancreatic acinar cells is mediated to a similar extent through both M_1_R and M_3_R receptors [11].

M_3_Rs, as other muscarinic receptors, are G-protein coupled receptors (GPCRs). Agonists that bind to GPCRs activate various signaling pathways within the cell. When bound to M_x_R, ACh produces a series of effects on intracellular lipid metabolism by stimulation of the activity of various phospholipases, yielding a series of fast and slow acting metabolites. The best characterized cytosolic signaling pathway is activation of phospholipase C (PLC) (Figure 1). PLC is activated to hydrolyze the phosphoinositide (PI) to produce 1,4,5-inositol trisphosphate (IP_3_) and diacylglycerol (DAG) [20,21,22,23]. PLC is additionally activated by the secretagogues, such as glucose, but also by depolarizing agents or cytosolic Ca^2+^ concentration ([Ca^2+^]_c_) elevation [24,25]. It can also be influenced through other GPCRs. For instance, glucagon-like peptide 1 (GLP-1) binds to a GPCR and strongly stimulates the activation of PLC [26]. DAG on the other hand activates PKC and subsequently sensitizes insulin exocytosis to Ca^2+^ ions [2,27,28].

Ca^2+^ ion concentration in the cell cytoplasm ([Ca^2+^]_c_) of both acinar and beta cells is kept low (10^−7^ M) and local or global oscillatory elevations in the concentration of these ions play a critical signaling role. Importantly, [Ca^2+^]_c_ elevation can occur in the absence of extracellular Ca^2+^, strongly suggesting that Ca^2+^ is stored and can be mobilized also from the intracellular compartments [29]. The Ca^2+^ concentration in extracellular space and in the internal stores is a few orders of magnitude higher than in the cytosol. Difference in Ca^2+^ concentration across the membranes is the major driving force for influx of Ca^2+^ ions, which runs through numerous transporters and Ca^2+^ release channels.

Since the early eighties, there is evidence that elevated IP_3_ levels in the cytosol trigger Ca^2+^ release from nonmitochondrial intracellular stores in acinar cells, the same stores that get released after ACh stimulation [30]. Soon thereafter, IP_3_ has been recognized as an important second messenger coupling membrane receptor activation with Ca^2+^ mobilization [31,32]. IP_3_ binds specifically to the inositol (1,4,5) trisphosphate receptor (IP_3_R) on the ER. Three isoforms of IP_3_R have been identified in mouse beta cells, type 1, type 2 and type 3, and the expression of the latter two is predominant [26]. In human beta cells, type 3 IP_3_R is expressed predominantly [33]. Pancreatic acinar cells predominantly express IP_3_R type 2 and type 3 [34]. Several important domains have been identified on the IP_3_Rs, with the ligand-binding, regulatory and channel domain [35]. The channel domain mediates the release of intracellular Ca^2+^ from ER to cytosol. In addition to the IP_3_ binding domain, IP_3_Rs have a Ca^2+^ binding domain. Ca^2+^ modulates the sensitivity of IP_3_R to IP_3_. IP_3_R cannot open and mediate the Ca^2+^ release from ER without Ca^2+^ binding beforehand. Importantly, Ca^2+^ alone is unable to activate IP_3_R [5].

Alongside IP_3_ and IP_3_Rs, additional second messengers and receptors responsible for Ca^2+^ mobilization have been identified [36]. After glucose stimulation, the secondary messenger cyclic adenosine 5′-diphosphate-ribose (cADPR) is produced. cADPR is known to increase Ca^2+^ mobilization through modulation of the ryanodine receptors (RYRs) [37]. Independently of IP_3_, the second messenger nicotinic acid adenine dinucleotide phosphate (NAADP) likewise acts on separate receptors to mobilize intracellular Ca^2+^ [38,39].

Oscillations in [Ca^2+^]_c_ are involved in the regulation of numerous cellular processes [41], including insulin secretion in beta cells [42]. [Ca^2+^]_c_ oscillations in beta cells were found to be of very different time scales, and are thought to correspond to pulsatile insulin release [43,44,45,46]. Similarly, Ca^2+^ signaling plays a prominent role for the digestive enzyme secretion in acinar cells [47,48]. However, the exact mechanisms that produce oscillations in [Ca^2+^]_c_ can vary significantly between the cells.

In acinar cells, intracellular Ca^2+^ stores have been well established to play a prominent role in generating Ca^2+^ oscillations. Acinar cells do not possess voltage-gated Ca^2+^ channels (VDCCs) [49], but do have store-operated Ca^2+^ influx from extracellular space [50]. There is accumulating evidence that the role of intracellular receptors and Ca^2+^ mobilization has been underestimated also in endocrine beta cells [51]. The current consensus model namely primarily builds on the key role of plasma membrane ion channels and resulting Ca^2+^ influx from extracellular space [52]. Briefly, glucose-dependent ATP production leads to a series of events to increase the opening probability of VDCCs, which mediate the Ca^2+^ influx into the cell and the resulting elevation of [Ca^2+^]_c_ triggers Ca^2+^-dependent exocytosis of insulin [26,45]. However, new experimental evidence, performed in more physiological conditions, supports a more central role for intracellular Ca^2+^ release channels as the dominant contributor to glucose-dependent Ca^2+^ oscillations and exocytosis of insulin. IP_3_ receptors, in addition to ryanodine receptors, were both shown to be sufficient and necessary to control [Ca^2+^]_c_ oscillations in beta cells [51]. On the other hand, Ca^2+^ influx through VDCCs may play only a secondary role in refilling the ER Ca^2+^ stores [51]. These novel insights call for the reassessment of the role of muscarinic activation and the IP_3_ signaling pathway in pancreatic beta cells.

Since the introduction of pancreas tissue slices [53], we have a technique that enables us to simultaneously study intact islet of Langerhans in the context of the surrounding acinar tissue. With this approach we can investigate preserved architecture and cellular function. Studying intact interaction under near-physiological conditions between homotypic and heterotypic cells is the biggest advantage tissue slices can offer [54]. Preserved environment of the endocrine and exocrine part of the pancreas is important to understand in vivo physiology and pathophysiology [55]. Pancreatic tissue slices have recently become a standard technique in a number of laboratories and have been successfully used for a broad spectrum of studies ranging from the characterization of islet microvasculature [56], analysis of diabetes pathogeneses [57], studying the function of pancreatic ductal cells [58], interaction between the different cell types [59], and Ca^2+^ dynamics [60]. The major aim of this paper has been to use the in situ character of the pancreatic slice preparation to learn about the cholinergic signaling by simultaneous observations of [Ca^2+^]_c_ dynamics in endocrine and acinar cells, in both physiological and supraphysiological conditions.

## 2. Materials and Methods

### 2.1. Ethics Statement

Study was conducted in accordance with national and European recommendations on care and handling experimental animals. We made all efforts to minimize the suffering of animals and to implement improvements in animal care and welfare. Administration of Republic of Slovenia for Food Safety, Veterinary and Plant Protection approved the experimental protocol (No: U34401-12/2015/3) and so did The Ministry of Education, Science and Research, Republic of Austria (No: 2020-0.488.800)

### 2.2. Tissue Slice Preparation and Dye Loading

Male and female C57BL/6J mice, 12–26 weeks of age, were kept on a 12:12 h light:dark schedule in individually ventilated cages (Allentown) at room temperature 22–24 °C and 45–55% relative humidity. Pancreas tissue slices were prepared as described before [53,61]. Briefly, mice were killed using CO_2_. Next, laparotomy was performed to access the abdominal cavity. Before the injection of the low-melting-point 1.9% agarose (Lonza, Basel, Switzerland), bile duct at the major duodenal papilla was clamped, allowing agarose to preferentially perfuse pancreas. Agarose was dissolved in extracellular solution composed of (in mM) 125 NaCl, 2.5 KCl, 26 NaHCO_3_, 1.25 NaH_2_PO_4_, 2 Na pyruvate, 0.25 ascorbic acid, 3 myo-inositol, 6 glucose, 1 MgCl_2_, 2 CaCl_2_ and 6 lactic acid (ECS). Temperature of injected agarose was 40 °C. After the injection, perfused pancreas was cooled with ice-cold ECS (ECS was buffered with CO_2_). Injection of agarose was a prerequisite for successfully conducted experiments. Ultimately, we cut 140 µM thick tissues slices using vibratome (VT 1000 S, Leica). Slices were transferred into 6 mM HEPES-buffered solution (HBS) consisting of (in mM) 150 NaCl, 10 HEPES, 5 KCl, 2 CaCl2, 1 MgCl2; titrated to pH = 7.4 using 1 M NaOH. For staining we used solution consist of 6 µM Calbryte 520, 0.03% Pluronic F-127 (*w/v*) and 0.12% dimethylsulphoxide (*v/v*) dissolved in HBS. Slices were stained with Calbryte 520, a Ca^2+^ (AAT Bioquest) for 50 min at room temperature. After staining, slices were transferred and kept in extracellular HEPES solution (in mM): 125 NaCl, 2.5 KCl, 10 HEPES, 10 NaHCO_3_, 1.25 NaH_2_PO_4_, 2 Na pyruvate, 0.25 ascorbic acid, 3 myo-inositol, 6 glucose, 1 MgCl_2_, 2 CaCl_2_, 6 lactic acid; titrated to pH = 7.4 using 1 M NaOH, until Ca^2+^ measurement. All chemicals, if not specify otherwise were obtained from Sigma-Aldrich, St. Louis, MO, USA.

### 2.3. Calcium Imaging

Imaging was performed on a Leica TCS SP5 upright confocal system using a Leica HCX APO L water immersion objective (20×, NA 1.0). Calbryte 520 was excited by a 488 nm argon laser. Emitted fluorescence was detected and measured by a Leica HyD hybrid detector in the range of 500–700 nm with the standard mode (Leica Microsystem, Wetzlar, Germany). Frequency of imaging was set to around 20 Hz at 256 × 256 pixels. Pixel size was close to 1 µm^2^, enabling quantification of Ca^2+^ oscillations.

The islet-containing slices that were subjected to imaging were chosen randomly to avoid biased selection from the head, neck, or tail of pancreas. Three to five slices were imaged from each mouse pancreas, and 6 or 8 mM glucose-containing, temperature-maintained (37 °C) ECS were perfused to the pancreas slice during Ca^2+^ imaging. Supraphysiological (25 µM) or the physiological (50 nM) concentrations of ACh were added in the ECS to study cholinergic effects. Time period of exposure to 8 mM glucose with or without ACh was adjusted depending on the time until the plateau activity was achieved. After the glucose stimulation, slices were perfused with substimulatory glucose (6 mM), before 8 mM glucose was reintroduced to the slices, or otherwise the experiments were discontinued.

### 2.4. Analysis

Analyses were performed as described before by Postić et al. [51]. In brief, we followed the analysis pipeline to first automatically detect ROIs (regions of interest) and sampled information about the time profile of [Ca^2+^]_c_ changes, their spatial coordinates, and neighboring ROIs, as well as about movie statistics, recording frequency, and pixel size. In the next step we distilled out all significant changes (z-score > 4) in [Ca^2+^]_c_ at all realistic time scales within each ROI. The events were characterized by the start time (t_0_), their maximal height, and the width at the half of their peak amplitude. The event was considered real if it was detected at multiple timescales and started around the same time and had approximately the same halfwidth. If their start and end times were within 20% of the halfwidth, they were considered cognate events, having arisen due to the same real event. Events that lasted less than three timeframes, as well as those too close to the beginning or end of the recording, were neglected. Figure 2a demonstrates the result of the automatic ROI selection and how different cell types were selected within the recorded part of the pancreatic slice (see Appendix A).

All the data samples that were used in statistical analysis failed the normality test. We used Mann-Whitney Rank Sum test to calculate the median, Q1 and Q3 as well as to assess the statistical significance. *p*-value smaller than 0.025 was considered as significant.

## 3. Results

In this study we used confocal microscopy with high spatial and temporal resolution to record, and custom-made Python code to identify individual cells and realistic time scale oscillations in [Ca^2+^]_c_ in pancreatic tissue slices. [Ca^2+^]_c_ oscillations were induced by elevation of physiological glucose concentration (8 mM) alone or in combination with either physiological (50 nM) or supraphysiological (25 µM) concentration of ACh. We particularly focused on M_x_R triggered intracellular Ca^2+^ release, which is well established in pancreatic acinar cells, and compared the responses in both cell types recorded at the same time. In this way we could directly compare the concentration dependence of the response to glucose and ACh in both cell types. The experiments were designed to validate the importance of ACh stimulation in metabolic homeostasis.

We intended to answer two major questions: (1) can we, in addition to morphological features, use the functional pattern of changes in [Ca^2+^]_c_ to distinguish the ACh responses in beta and acinar cells? (2) Can the improved spatially and temporally resolved imaging of [Ca^2+^]_c_ oscillations and optimized data analysis yield new insights into the mechanisms of ACh actions in pancreas?

### 3.1. Morphological and Functional Determination of Cell Types in Pancreatic Tissue Slices

Being able to distinguish between the different cell types in a composite tissue is a prerequisite for the analysis of temporal changes in the physiological parameters. Islets in a pancreas of wild-type mice are mostly well defined interlobular structures, which can be easily morphologically separated from the neighboring acinar cells (Figure 2a) [7]. Islet composition in mouse islets has been described to have beta cells occupying the majority of the central part of the islet, while non-beta cells form a mantle part, with important exceptions to the rule [7]. Furthermore, a typical biphasic activity after exposure to stimulatory glucose concentration is a distinctive characteristic of active beta cells (Figure 2b) [61]. For this study, we selected and further analyzed ROIs that represent beta cells and functional regions of acinar cells, based on the morphological features and their [Ca^2+^]_c_ oscillations pattern (Figure 2). Physiological [Ca^2+^]_c_ elevations in acinar cells occur locally in the apical granular pole [50]. Automatic detection of ROIs in our analyzing system have been optimized for detecting islet cells, which are compared to acinar cells smaller in size. Therefore, automatically detected ROIs in acinus region do not correspond to the size of acinar cell, but to the locally functioning region of acinar cells.

At the substimulatory 6 mM glucose concentration, beta cells had stable resting [Ca^2+^]_c_ (Figure 2b). Physiological stimulatory concentration of glucose, after a delay of several hundred seconds, triggered a biphasic Ca^2+^ response, composed of a series of slow, non-synchronized oscillations in [Ca^2+^]_c_ during the initial transient phase (Figure 2b). This delay can differ quite substantially between the islets when stimulated with physiological glucose concentration. The initial phase was followed by prolonged plateau activity with fast synchronized oscillations (Figure 2b). This plateau activity diminished reliably when glucose was lowered back to 6 mM (Figure 2b). Note that the initial beta cell response to stimulatory glucose concentrations is very heterogenous with different onset times, different sizes of activating cell clusters, and duration as well as number of slow oscillatory events, which is in line with our previous work [61]. Slower oscillations in the transient phase have a mean duration of some tens of seconds (Figure 2c,d). On the other hand, during the plateau phase, well synchronized faster events with a duration of 2–3 s predominate. Those events are, however, also compound events, presenting a temporal summation of even faster Ca^2+^ spikes that are believed to correspond to burst of membrane potential depolarizations.

Acinar cells in situ reproducibly responded to glucose stimulation with slow [Ca^2+^]_c_ oscillations that were mostly not synchronized between the cells. The dominant halfwidth duration of these oscillations was above 10 s (Figure 2 and Figure 3). The glucose response was concentration-dependent, with a faster onset to 8 mM glucose stimulation and delayed switch off during the substimulatory 6 mM glucose washout period compared to beta cells (Figure 2 and Figure 3).

Our results showed that based on the morphological and functional features, after stimulation with a physiological glucose concentration, it is possible to reliably differentiate pancreatic beta cells from acinar cells by measuring [Ca^2+^]_c_ changes in situ. The [Ca^2+^]_c_ oscillations in beta cells are about an order of magnitude more frequent compared to acinar cells and are by about the same factor shorter in duration, but also well-synchronized. In acinar cells, physiological levels of ACh enhanced synchronization and shortened the duration of the events (Figure 2e,j).

### 3.2. Differential Effects of Glucose and ACh on Oscillations in [Ca^2+^]_c_

In the second part of the experiments, ACh was coapplied during the 8 mM glucose stimulation (Figure 2 and Figure 3, Appendix A). Under these stimulatory conditions, the application of a physiological concentration of ACh still produced an apparent biphasic response in beta cells. However, the onset time delays between different cell clusters during the transient phase were diminished (Figure 2b,d) to the extent that the typical time profile of the compound slow events we could record from individual ROI representing a beta cell became apparent also on the average signal representing all active beta cells (Figure 2g). The median delay to the first event was reduced from 358 s (Q1 308 s, Q3 576 s) to 223 s (Q1 182 s, Q3 253 s, *p* < 0001). Furthermore, this stimulation produced an elevation of the frequency of the fast [Ca^2+^]_c_ oscillatory activity during the plateau phase and the halfwidth duration of the oscillations decreased from 1.6 s (Q1 1.3 s, Q3 1.9 s) to 1.3 s (Q1 1.1 s, Q3 1.6 s, *p* < 0.001) (Figure 2c). Similar to faster and synchronous onsets of [Ca^2+^]_c_ oscillations, these events also tended to wash out faster when the stimulatory glucose concentration was replaced with a non-stimulatory 6 mM glucose in the ECS.

In acinar cells, addition of 50 nM ACh triggered [Ca^2+^]_c_ oscillations, which were better synchronized between the ROIs compared to glucose stimulation alone (Figure 2i) as can be seen from the average signal from all functional regions of acinar cells (Figure 2j). The events with the median halfwidth value of about 7, did not change after the ACh stimulation (Figure 3e). The peaks of [Ca^2+^]_c_ recorded at this ACh concentration showed only moderate temporal summation, which would confirm that we stimulated in the physiological range (Figure 2b,i,j). Acinar cells responded to ACh sooner (median 140 s, Q1 104 s, Q3 179 s) compared to the activation onset of beta cells in the same slice (median 223 s, Q1 181 s, Q3 253 s, *p* < 0.001).

Supraphysiological concentrations of ACh have been more often used in isolated preparations of pancreas tissue, such as isolated islets or isolated acini. In our experiments, both beta and acinar cells responded to 25 µM ACh with a biphasic response (Figure 3b,g). At this high ACh concentration, the initial response was still faster in acinar cells (median 43 s, Q1 42 s, Q3 44 s) compared to beta cells (median 47 s, Q1 46 s, Q3 48 s, *p* < 0.001) (Figure 3d,i); however, the difference was only a few seconds. At the level of individual beta cells, the second phase was characterized with higher frequency oscillations compared to glucose stimulation alone, and with a shorter dominant halfwidth duration of [Ca^2+^]_c_ events (median 2.8 s, Q1 1.9 s, Q3 3.8 s) compared to glucose only (median 3.0 s, Q1 1.8 s, Q3 4.2 s, *p* < 0.025) (Figure 3c). The major difference to the glucose only control conditions or physiological ACh stimulation was that the fast plateau oscillations in beta cells were no longer synchronized on the islet level (Figure 3c,g). On the temporal profile showing an average of all active beta cells, oscillations are therefore hardly noticeable (Figure 3h). Interestingly, the beta cells resynchronized rapidly after ACh has been washed out of the recording chamber (Figure 3b,h). This result showed that the high ACh effect washed out faster than that of stimulatory glucose (Figure 3b,d).

In acinar cells, the second phase of the response to 8 mM glucose and 25 µM ACh produced a sustained and elevated plateau, with only weakly detectable oscillations in [Ca^2+^]_c_ (Figure 3f,i). Elevation in [Ca^2+^]_c_ under these conditions is pathologically high, which has also been picked up by the size of the ROIs, which now covered the complete area of the acinar cell, showing activity in apical as well as basolateral pole of an acinar cell.

Our results show that ACh stimulation produced concentration-dependent dual effects and that ACh above micromolar concentration does not represent physiological conditions.

## 4. Discussion

The morphological identification of exocrine acinar cells in wild-type mouse pancreatic tissue slices has been well established and straightforward. Our study here provides additional means to distinguish acinar cells from islet beta cells. We revealed the different functional features between these two cell types upon stimulation with secretagogues with known effects. All the more valuable can this functional discrimination become in discrimination of endocrine cells from acinar cells in studying human tissue slices, slices of the diabetic or genetically modified mouse models, where the morphological features are not always apparent.

In acinar cells, stimulation of M_1_Rs and M_3_Rs, and the ensuing formation of IP_3_ is the major mechanism to robustly driving digestive enzyme release [11]. M_3_Rs were suggested to play a key role in maintaining insulin release and nutrient homeostasis of beta cells [19]. However, the exact mechanisms downstream of these receptors activation in beta cells are less well known, since a number of different mechanism have been discussed [2]. Even more controversial is the role of glucose-dependent stimulation on acinar cells which should, at least transiently, play an important physiological role in stimulation of the amylase release. In this study we tried to correlate the effects of stimulatory glucose and ACh measured simultaneously from the fresh pancreatic tissue slice.

In a number of previous studies, ACh was used predominantly in its supraphysiological range to determine the effect of this parasympathetic neurotransmitter on the islets electrical activity and [Ca^2+^]_c_ oscillations [2]. ACh in supraphysiological concentrations was used to study cholinergic signal transduction in several other tissues and cell types [62,63]. Glucose and ACh were found to utilize secretagogue-specific effects on the electrical activity pattern [64]. Increasing glucose concentration resulted in burst prolongation and eventually reduced frequency [46]. On the other hand, higher ACh increased frequency, while the burst duration did not increase [64].

At stimulatory glucose concentrations, addition of physiological concentration of ACh should primarily trigger membrane depolarization mediated by a Na^+^-dependent mechanism, which was suggested to operate through transient receptor potential-canonical (TRPC) channels [65]. This depolarization further induces Ca^2+^ influx through VACCs, which is compatible with an increased frequency of [Ca^2+^]_c_ events. All effects of ACh on membrane potential could be antagonized by atropine, suggesting an involvement of M_x_Rs [64]. At pharmacological ACh concentrations, the primary mechanism should involve Ca^2+^ mobilization from the ER and inhibition of L-type VACCs. The latter was suggested to serve prevention of a deleterious Ca^2+^ overload [2]. Interestingly, despite the inhibition of VACCs at high ACh, insulin release was not inhibited unless stimulation was prolonged [2].

What new can we learn from our analysis? Both glucose and ACh in the physiological range stimulate [Ca^2+^]_c_ oscillations in beta and acinar cells. Acinar cells are more sensitive to both stimulations compared to beta cells. They were shown to switch on earlier and switch off later. In case of glucose stimulation this likely reflects differential expression of GLUT transporters in these two different cell types. Beta cells in mice express low-affinity GLUT2 transporters, while acinar cells express high-affinity GLUT1 and GLUT3 transporters [66]. Higher sensitivity to ACh in acinar cells may on the other hand reflect scarcer innervation of exocrine tissue compared to the innervation of islets [7]. At saturating concentrations of ACh, both cell types activated practically at the same time.

One of the more prominent effects of physiological ACh stimulation is complete synchronization of the activation onsets between different beta cells in an islet. A possible explanation is that in the slices there is a residual tonus of ACh related to the innervation of individual beta cells, which contributes to apparent heterogeneity between the activation onsets in different beta cells. Addition of a physiological level of ACh possibly overrides these local differences. Alternatively, pancreatic tissue slices are thick enough to contain intraganglionic neurons that express nicotinic ACh receptors. Future work with antagonists of both muscarinic and nicotinic receptors is needed to resolve this issue. At more extreme ACh concentrations, the frequency of [Ca^2+^]_c_ oscillations in individual beta cells was further increased; however, the synchronization of the events on the plateau phase between beta cells in an islet was completely lost. It has been previously reported that fast [Ca^2+^]_c_ oscillations during the plateau phase and their synchronization predominantly reflect the activity of RYRs [51]. An enhanced stimulation of IP_3_ receptors due to chronic muscarinic stimulation and IP_3_ production could interfere with the function of RYR receptors and lead to desynchronization.

Ca^2+^ oscillations mediated by muscarinic activation depend on agonist concentration and we believe they are essential for not only physiological response of beta cells to glucose, but also normally functional acinar cells. Furthermore, elevated intracellular Ca^2+^ alone cannot promote IP_3_ production in beta cells [51], which is another piece of evidence to support the importance of ACh signaling through muscarinic activation. Moreover, IP_3_R activation could be evoked with ACh in combination with substimulatory glucose concentration and is resulting in [Ca^2+^]_c_ oscillations very akin to oscillation observed in the first phase during the 8 mM glucose stimulation. In addition, significance of the IP_3_ signaling pathway was shown with the IP_3_ inhibitor Xestospongin C, which selectively blocks activity of IP_3_. Xestospongin C blocked events in the transient phase and changed the phenotype of the first initial phase, but left the plateau phase intact [51]. It is well established that ACh in acinar cells mediates action through the muscarinic receptors, followed by IP_3_ production that consequently generates Ca^2+^ oscillations [67], hence ACh stimulates acinar enzyme secretion [68]. Experimental data, obtained from isolated single acinar cells with specific muscarinic receptor knock-out, strongly suggest the importance of muscarinic activation and IP_3_ signaling in acinar cells [69].

Acinar cells at physiological concentrations of agents such as ACh, that evoke Ca^2+^ from intracellular stores, start local and short lasting [Ca^2+^]_c_ oscillations, which are synchronized among the neighboring cells, and which control enzyme and acinar fluid secretion [70]. High supraphysiological ACh concentrations generated a distinct Ca^2+^ spike followed by abolished [Ca^2+^]_c_ oscillations. The described course of events in acinar cells is expected when [Ca^2+^]_c_ is drastically increased, eventually resulting in long-term inhibition of digestive enzyme secretion [71].

In summary, fresh tissue slices are the method of choice to simultaneously assess different endocrine and exocrine cell types in pancreas. Both morphological and progressively better functional characterization can provide useful information about differences between the observed cell types as well uncover delicate interactions between these cell types. The high sensitivity to physiological and pharmacological substances better represents the in vivo situation compared to dispersed cells or cell clusters. Our findings provide an important base for the future studies in physiology and pathophysiology of pancreas using mouse models and tissue from human donors. However, beta cell function can be inherently tied to calcium activity and caution is required to identify dysfunctional beta cells in severe phenotypes.

## Figures and Tables

**Figure 1 cells-10-01580-f001:**
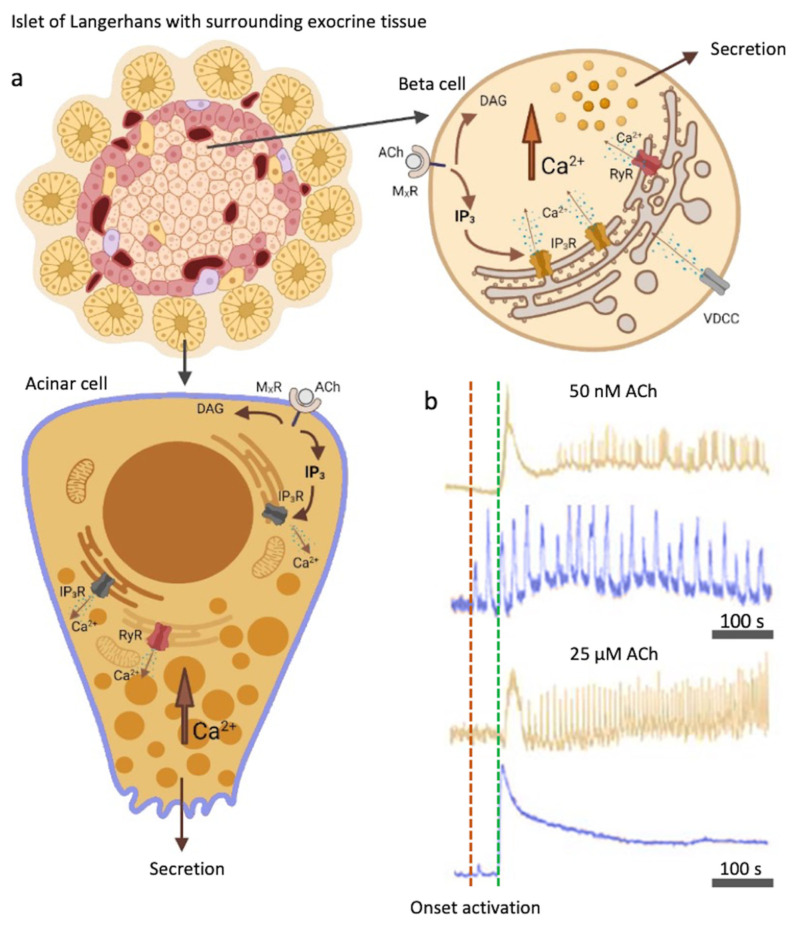
Muscarinic activation with ACh and the IP_3_ signaling pathway in beta cells and acinar cells. (**a**) Acinar and beta cells express muscarinic receptors (M_X_Rs), activation of which results in IP_3_ and DAG production. IP_3_ specifically binds to IP_3_R on the ER and triggers Ca^2+^ mobilization, leading to an oscillatory rise in [Ca^2+^]_c_. Additionally, RYR receptors can be activated. L-type VDCCs play a role in Ca^2+^ refilling of the ER. (**b**) Color-coded time courses for beta (light brown) and acinar cells (blue) simultaneously stimulated with 8 mM glucose in combination with a physiological (50 nM, top traces) or a supraphysiological (25 µM, bottom traces) ACh concentration. The activation onsets after the cells were stimulated for the acinar cells (red) and the beta cells (green) are indicated with the vertical dashed lines. Legend: M_x_R = muscarinic receptor, ACh = acetylcholine, DAG = diacylglycerol, IP_3_ = 1,4,5-inositol trisphosphate, IP_3_R = inositol (1,4,5) trisphosphate receptor, RYR = ryanodine receptors. Created with [40].

**Figure 2 cells-10-01580-f002:**
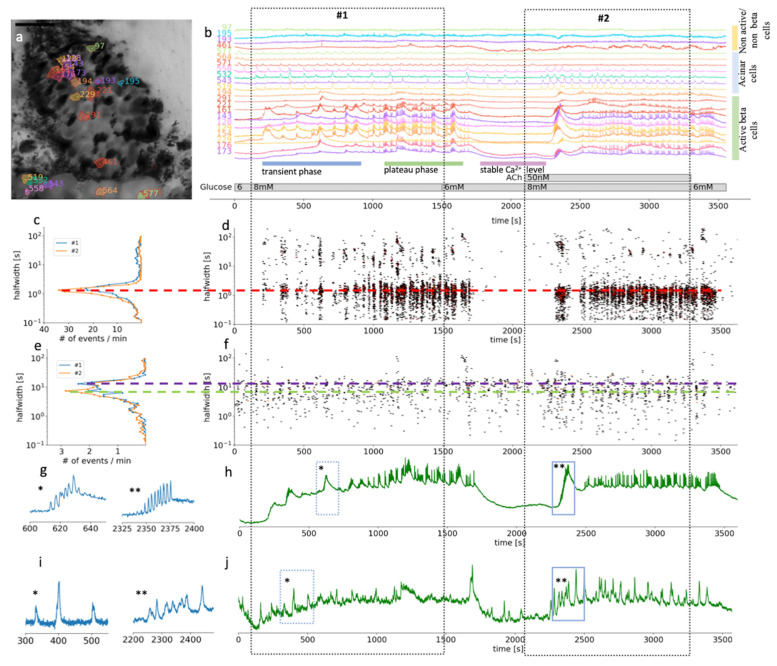
Time scales and heterogeneity of responses of cytosolic Ca^2+^ oscillations in mouse beta and acinar cells stimulated with 8 mM glucose or 8 mM glucose plus 50 nM ACh. (**a**) Localization of several beta, acinar and non-active beta on non-beta cells in a pancreas tissue slice. (**b**) Time course of the [Ca^2+^]_c_ changes in beta, acinar and non-active beta on non-beta cells selected in (**a**). The two dashed rectangles stretching across the panels indicate the periods of stimulation. (**c**) Frequency histogram of the halfwidth durations of the events and (**d**), onset time of the [Ca^2+^]_c_ events at measured time scales for beta cells (*n* = 82). Addition of 50 nM ACh increased the frequency of the [Ca^2+^]_c_ events, the dominant halfwidth duration stayed unchanged (red dashed line). (**e**) Frequency histogram of the halfwidth durations of the events and (**f**), onset time of the [Ca^2+^]_c_ events at measured time scales for functional region of acinar cells (*n* = 48). Addition of 50 nM ACh increased the frequency of the [Ca^2+^]_c_ events, the dominant halfwidth length shortened (blue and green dashed line). (**g**) Expanded time traces of an individual ROI (*) from a representative beta cells or an average of all ROIs (**) of active beta cells (*n* = 82) as indicated by small blue rectangles in (**h**). (**i**) Expanded time traces of an individual ROI (*) from a representative acinar ROI or an average of all functional regions of acinar cells ROIs (**) (c = 48) as indicated on the temporal profile in (**j**) with small blue rectangles.

**Figure 3 cells-10-01580-f003:**
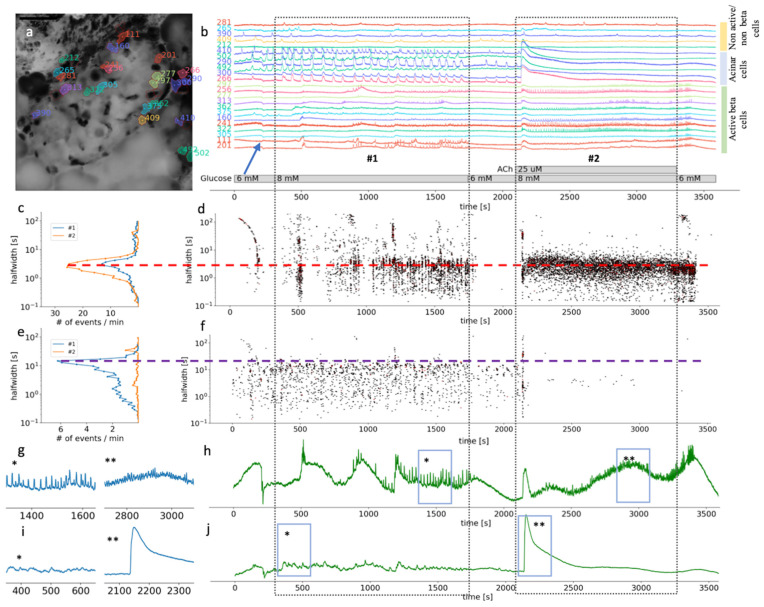
Time scales and heterogeneity of responses of cytosolic Ca^2+^ oscillations in mouse beta and acinar cells stimulated with 8 mM glucose or 8 mM glucose plus 25 µM ACh. (**a**) Localization of several beta, acinar and non-active beta on non-beta cell in pancreas tissue slices. (**b**) Temporal profile of the [Ca^2+^]_c_ changes in a beta, acinar and non-active beta on non-beta cells shown in pancreas tissue slice in (**a**). The two dashed rectangles stretching across the panes indicate the periods of stimulation. (**c**) The frequency histogram of the halfwidth durations of the events and (**d**), onset time of the [Ca^2+^]_c_ events at measured time scales for active beta cells (*n* = 113). Addition of 25 μM ACh increased the frequency of the [Ca^2+^]_c_ events, the dominant halfwidth duration stayed unchanged (red dashed line). (**e**) The frequency histogram of the halfwidth durations of the events in acinar cell and (**f**), onset time of the [Ca^2+^]_c_ events at measured time scales for functional regions of acinar cells (*n* = 33). Addition of 25 μM ACh abolished [Ca^2+^]_c_ oscillations of the dominant scale (violet dashed line). (**g**) Expanded time traces of an average from all ROIs specify as active beta cell shown in (**h**) (*n* = 113). (**h**) (*) is indicating plateau phase of the response to glucose stimulatory concentration. (**) is indicating fast events in plateau phase during stimulation with 25 µM on top of the 8 mM glucose in panel. (**i**) Expanded time traces of an average from all ROIs correspond to functional regions of acinar cells (*n* = 33), shown in (**j**). (*) is indicating average [Ca^2+^]_c_ events during stimulation with 8 mM glucose. (**) is indicating average [Ca^2+^]_c_ events during stimulation with 25 µM ACh on top of the stimulatory glucose concentration. Blue arrow on (**b**) is indicating an artefact due to a transient slice movement that occurred during imaging and was detectable in multiple time traces.

## Data Availability

The data presented in this study are available on request from the corresponding author. The data are not publicly available due to their large size.

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
