# Peer review of "Dual Mode of Action of Acetylcholine on Cytosolic Calcium Oscillations in Pancreatic Beta and Acinar Cells In Situ"

_cells, 2021, doi:10.3390/cells10071580_

Round 1

Reviewer 1 Report

The authors describe the activation of acinar and beta-cells in situ in pancreatic tissue slides by glucose and acetylcholine, demonstrating that these cell types exhibit unique responses and can be differentiated by their oscillatory calcium response. The authors conclude that these changes are likely due to differential glucose transporter expression. The results reported appear scientifically sound though not groundbreaking, and these findings may useful for those in the islet biology field. Edits to data presentation  help with clarity, and additional discussion regarding mechanism and biological relevance would improve the manuscript.

Main comments:

Figure 1 does not have an in-text reference.

Figure 1(b) would benefit from a more contrasting colour in the traces of the two cell types. From the Figure Legend, it is unclear the significance of why the stimulation indicators (presumably dotted blue and green lines) do not match up between the two cells lines.

Mouse islet architecture (also indicated in Figure 1) usually dictates that beta-cells are situated in the core of the islet structure, whereas alpha-cells form a mantle. Though the authors mention using ‘morphology’ to select cells, they should add more information in their analyses as to how they excluded non-beta cells from their experiments and are able to confidently ascertain the measurement taken were from beta-cells.

Figure 2 and 3: Though acinar cells and beta-cells can be differentiated by their traces, it would add clarity to the data if the traces were more distinctly grouped in the Figure by cell type.

Presumably due to the unmarked scale of the traces in figure 2b and Figure 3b, it is difficult to compare even the basal conditions (6mM and 8mM glucose). Visually, the beta-cells in Figure 3 appear much less responsive. It may be beneficial to use the same scale for both Figure 2b and 3b, or include a representative trace of acinar cells and beta-cells from each experiment instead.

As a major conclusion of the study is that physiological ACh concentrations activate synchronisation of beta-cells unlike glucose-alone, one may wonder if the beta-cells are “pre-synchronised” from the 8mM glucose-only plateau phase prior to 50nM ACh stimulation, explaining the synchronous activation of all beta-cells. It is therefore a necessary control experiment to have the authors stimulate beta-cells/acinar cells with ACh without prior glucose-only conditions.

The authors should comment on the presence of "hub" beta-cells (PMID 30142036, PMID 27452146) that coordinate islet calcium oscillations. This may be what is being observed in initial transient phase heterogeneity, with beta-cells synchronising in the plateau phase.

The authors should comment on the use of 25uM ACh as a pharmacological dose, considering their results in Figure 3 indicate that these are not normal physiological conditions. Has this dose been used in other tissue types? Are there any situations in which pancreatic tissue observe this range of ACh?

The authors also note that these methodologies will be valuable in the discrimination of models of diabetes or transgenic mice, where morphological features are hard to distinguish. However, Is it known whether these calcium responses may be affected by diabetes (or genetic models that may affect islet morphology)? Beta-cell function is inherently tied to calcium activity, and it must be acknowledged that it may not be useful to use these measurements to identify dysfunctional beta-cells.

Reviewer 2 Report

In this paper from Sluga N et al, the authors show a method for separating beta cells from acinar cells depending on cytosolic calcium oscillations in response to Acetyl-choline plus minus glucose. Although the paper is well designed and performed, several concerns have arisen after reading carefully the manuscript:

  1. The authors show different experiments with 8 mM of glucose concentration. However, i would like to know what would happen when the experiments is made with 15-16 mM, which are related with insulin secretion by pancreatic beta cells.
  2. To corroborate the results obtained of glucose 8 mM +/- Acetylcholine in calcium oscillations, the authors should present the results using an alternative technique, such as patch-clamp 
  3. The authors must utilize an inhibitor of of acetylcholine in order to revert its effect in glucose action on calcium oscillations

Round 2

Reviewer 1 Report

I am satisfied with the authors' response to my comments and can recommend the manuscript for publication. 

Reviewer 2 Report

All the questions and comments arisen by this referee have been answered adequately.